# In Vitro Therapeutic Efficacy of Furazolidone for Antimicrobial Susceptibility Testing on *Campylobacter*

**DOI:** 10.3390/antibiotics14070636

**Published:** 2025-06-22

**Authors:** Jeel Moya-Salazar, Alfonso Terán-Vásquez, Richard Salazar-Hernandez, Víctor Rojas-Zumaran, Eliane A. Goicochea-Palomino, Marcia M. Moya-Salazar, Hans Contreras-Pulache

**Affiliations:** 1Faculty of Medicine, Universidad Señor de Sipán, Chiclayo 14002, Peru; 2Department of Pathology, Hospital Nacional Docente Madre-Niño San Bartolomé, Lima 51001, Peru; alfonsoter231@gmail.com (A.T.-V.); rovicvikthor@gmail.com (V.R.-Z.); 3Department of Pathology, Hospital Nacional Guillermo Almenara Irigoyen, Lima 15001, Peru; richard.sh30@gmail.com; 4Faculties of Health Science, Universidad Tecnológica del Perú, Lima 51001, Peru; elagoichi@gmail.com; 5Medical Unit, Nesh Hubbs, Lima 51001, Peru; marciamilagros21@gmail.com; 6Faculty of Medicine, Universidad Norbert Wiener, Lima 51001, Peru

**Keywords:** *Campylobacter*, antimicrobial resistance, furazolidone, pediatrics, fluoroquinolones, multidrug resistance

## Abstract

**Background:** *Campylobacter* causes gastroenteritis worldwide with increasing antimicrobial resistance. Furazolidone (FZD) shows potential in resource-poor areas but needs further study. We aimed to assess the in vitro susceptibility of *Campylobacter* spp. to FZD, ciprofloxacin (CIP), and erythromycin (ERY) in a high-risk pediatric cohort and to evaluate the clinical relevance of resistance patterns using inhibitory quotient (IQ) pharmacodynamics. **Methods:** A two-phase prospective study (2012–2013, 2014–2015) was conducted at a tertiary pediatric hospital in Lima, Peru. Stool samples from children ≤24 months were cultured on selective media, with *Campylobacter* isolates identified via conventional bacteriological methods. Antimicrobial susceptibility was determined using Kirby–Bauer disk diffusion and regression-derived minimum inhibitory concentrations (MICs). IQ analysis correlated inhibition zones with therapeutic outcomes. **Results:** Among 194 *Campylobacter* isolates (*C. jejuni*: 28%; *C. coli*: 72%), resistance to CIP declined from 97.7% (2012–2013) to 83% (2014–2015), while ERY resistance rose from 2.3% to 9.4% (*p*= 0.002). No FZD resistance was observed, with mean inhibition zones of 52 ± 8 mm (2012–2013) and 43 ± 10.5 mm (2014–2015). MICs for FZD were predominantly <0.125 μg/mL, and all susceptible isolates demonstrated favorable IQ outcomes. Multidrug resistance (≥2 drugs) increased to 6.2% (2014–2015), though all MDR strains retained FZD susceptibility. CLSI and EUCAST breakpoints showed concordance for ERY (*p* = 0.724) but discordance for CIP (*p* = 0.022 vs. 0.008). **Conclusions:** FZD exhibits sustained in vitro efficacy against *Campylobacter* spp., even among MDR strains, contrasting with escalating fluoroquinolone and macrolide resistance.

## 1. Introduction

*Campylobacter* infection is a leading cause of gastroenteritis worldwide, with significant public health and economic implications. In the United States alone, the incidence rate of campylobacteriosis is 14.3 cases per 100,000 individuals, resulting in an estimated annual economic burden of USD 1.7 billion [1,2]. Among the *Campylobacter* species, *Campylobacter jejuni subsp. jejuni* (*C. jejuni*) accounts for the majority of infections (95–98%), while *C. coli* is responsible for approximately 2–5% of cases [3]. Other species, such as *C. lari*, are rarely isolated from stool specimens of patients with gastroenteritis, while *C. fetus subsp.* fetus (*C. fetus*) is occasionally associated with systemic infections, including septicemia, particularly in immunocompromised individuals.

Although most cases of *Campylobacter* gastroenteritis are self-limiting, severe infections or those occurring in immunocompromised patients often require targeted antibiotic therapy. Erythromycin and ciprofloxacin remain the antimicrobial agents of choice for treatment [3]. However, the rising prevalence of antibiotic resistance in *Campylobacter* spp., particularly to fluoroquinolones, tetracyclines, and macrolides, has complicated therapeutic decision-making [4]. This trend underscores the urgent need for reliable methods to assess antibiotic susceptibility and guide empiric therapy.

The World Health Organization (WHO) has identified fluoroquinolone-resistant *Campylobacter* spp. as a high-priority pathogen, emphasizing the critical need for novel antibiotics [5,6]. This alarming situation highlights the potential consequences of a post-antibiotic era, where the emergence of multidrug-resistant “superbugs” could severely impact global public health, particularly in low- and middle-income countries (LMICs) [7].

In Peru, for example, the persistence of resistance to erythromycin and ciprofloxacin has led some pediatricians to empirically treat campylobacteriosis with nitrofurans, such as furazolidone, despite variable efficacy and the lack of formal recommendations from the Peruvian Directorate-General of Medicines, Supplies, and Drugs (DIGEMID) [8]. This empirical approach reflects the urgent need for effective therapeutic options, particularly in pediatric populations, which exhibit high rates of *Campylobacter* infection and limited access to safe, effective treatments. Ensuring appropriate management and monitoring of this disease in vulnerable populations is critical, as the available therapeutic arsenal is increasingly constrained by resistance and the risk of adverse effects [9].

We aimed to evaluate the therapeutic efficacy, in vitro, of furozolidone for antimicrobial susceptibility testing of *Campylobacter* spp., *C. jejuni*, and *C. non-jejuni*.

## 2. Results

### 2.1. First Study Phase (2012–2013)

A total of 88 patients were included, with *Campylobacter* isolates tested for antibiotic susceptibility. Resistance to FZD was absent (0/88), whereas 98% (86/88) of isolates exhibited resistance to CIP (MIC ≥ 4 μg/mL). Dual resistance to ERY and CIP was observed in 2.3% (2/88) of isolates. All ERY- and CIP-resistant strains correlated with unfavorable therapeutic responses. In contrast, FZD demonstrated global efficacy, with a mean inhibition zone diameter of 52 ± 8 mm (range: 38–65 mm; 95% CI: 50.2–53.8).

Comparative analysis of CLSI and EUCAST breakpoints revealed no significant discordance in resistance categorization (*p* = 0.724). Resistance rates were as follows: ERY (2.3%, 2/88), CIP (98%, 86/88), and FZD (0%, 0/88). All patients (mean age: 9 ± 1 month) were symptomatic and received targeted therapy post-microbiological confirmation.

### 2.2. Second Study Phase (2014–2015)

Out of 591 suspected bacterial infections, 485 (82%) were excluded as not being *Campylobacter*. Of the 106 (18%) isolates examined, *C. coli* was the most common, representing 72% (76/106), while *C. jejuni* made up 28% (30/106). Susceptibility profiles by species are detailed in Table 1.

No cases of antimicrobial resistance to FZD were reported (0/106). The mean inhibition zone diameter for FZD was 43 ± 10.5 mm (range: 15–70 mm; 95% CI: 41–45), and all strains were associated with favorable therapeutic outcomes. Resistance to CIP and ERY was detected in 81% (86/106) and 9.3% (10/106) of isolates, respectively. Only five (4.7%) isolates were resistant to ERY and CIP. The cohort (22% female; mean age: 9 ± 5 months; 95% CI: 8.1–10) exhibited comparable demographic trends to the first phase.

### 2.3. MIC and Inhibitory Quotient (IQ) Analysis

FZD MIC distributions were consistent across study periods (*p* = 0.817), with most isolates exhibiting MICs < 0.125 μg/mL and limited intermediate phenotypes (Figure 1). ERY MICs demonstrated significant divergence between susceptible (<1 μg/mL), intermediate, and resistant categories (*p* = 0.001). For CIP, 94% of isolates were susceptible (MIC: ≤36 μg/mL), 2% intermediate, and 12% resistant (≥36 μg/mL). Discordance between MICs and disk diffusion results was observed for CIP, with significant differences between CLSI (*p* = 0.022) and EUCAST (*p* = 0.008) interpretations.

CLSI-defined resistant strains for ERY and CIP globally correlated with unfavorable IQ-based therapeutic responses. Conversely, all susceptible strains (ERY, CIP, FZD) aligned with favorable IQ outcomes. Intermediate CLSI categorization for CIP (3.1%, 6/194) and ERY (5.2%, 10/194) corresponded to unfavorable responses. Similarly, all FZD-intermediate isolates showed unfavorable IQ outcomes.

### 2.4. Multidrug Resistance Trends

CIP resistance declined from 97.7% (2012) to 83% (2015), while ERY resistance increased from 2.3% to 9.4% (*p* = 0.002). No FZD resistance was detected in either period, and no significant difference was found among *Campylobacter* isolates during the years of study (*p* = 0.808).

Resistance to ≥1 drug was observed in 83.5% (162/194) of isolates, with 6.2% (12/194) resistant to ≥2 drugs (predominantly CIP + ERY). The most common multidrug resistance (i.e., ≥2 drugs) pattern included CIP and ERY. No isolates exhibited resistance to ≥3 drugs. Multidrug-resistant strains were exclusively isolated from infants < 1 year old, all retaining FZD susceptibility. The remaining antimicrobial drug had no statistically significant change in resistance over time (*p* = 0.948). Antimicrobial susceptibility patterns and species differentiation via hippurate hydrolysis are shown in Figure A1 and Figure A2.

## 3. Discussion

Our results show that FZD has promising in vitro therapeutic efficacy against *Campylobacter* species isolated from infants and children of Peru, with no resistance observed across both study phases. All FZD-susceptible isolates exhibited favorable IQ outcomes, whereas intermediate susceptibility correlated with unfavorable responses. Multidrug resistance (≥2 drugs) was observed in around 6% of isolates, predominantly CIP + ERY, but all remained FZD-susceptible.

### 3.1. Strengths

To the best of the author’s knowledge, this is the first Latin American in vitro study evaluating FDZ as a valuable option for antimicrobial therapy against *Campylobacter* infections. FDZ is recognized as effective against a variety of bacterial infections (i.e., *Helicobacter pylori, Vibrio cholerae*) due to its nitrofuran derivative properties [10,11]. It is often used in combination with other antimicrobials for enhanced effectiveness, especially in regions with high resistance to other antibiotics [12]. In Peru during the 1980s, a clinical trial was conducted for the treatment of gastritis and ulcers when *Helicobacter pylori* was still referred to as *Campylobacter pylori* [13], however, the efficacy of FZS against *Campylobacter* has not been previously reported. Another strength is working with one of the most worrisome bacteria worldwide. The WHO’s classification of fluoroquinolone-resistant *Campylobacter* as a high-priority pathogen underscores its threat to food safety and antimicrobial efficacy [6]. Our findings support the inclusion of nitrofurans in combinatorial therapies in settings with *Campylobacter* strain high-resistance antibiotics [14].

### 3.2. Main Discussion

Our finding aligns with historical reports of FZD’s efficacy against Campylobacter, particularly for *C. pylori*, and its use in eradicating *Helicobacter pylori* through urease suppression [15,16,17]. Mechanistically, FZD’s nitrofuran structure generates reactive intermediates that disrupt bacterial DNA and metabolic pathways and involves the enzymatic reduction in the parent compound to electrophilic radicals [12], a mode of action less prone to resistance compared to fluoroquinolones or macrolides. Notably, our findings contrast sharply with the near-ubiquitous resistance to CIP (98% in 2012, 81% in 2015) and rising ERY resistance (2.3% to 9.4%), reinforcing FZD’s potential as a salvage therapy.

The global favorable therapeutic response observed in FZD-susceptible isolates suggests consistent bactericidal activity. However, isolates categorized as “intermediate” by CLSI disk diffusion showed unfavorable IQ outcomes, likely reflecting suboptimal drug concentrations at infection sites. This discordance highlights the need for species-specific FZD breakpoints, as current guidelines do not provide breakpoints for FZD in stool cultures for *Campylobacter* [18,19]. Further research is necessary to avoid extrapolating nitrofuran breakpoints in the assessment of FZD susceptibility.

On the other hand, CIP resistance in *Campylobacter* infections is a growing concern worldwide. The alarming prevalence of CIP resistance (97.7% in 2012, 83% in 2015) is supported by several investigations that show a notably high *Campylobacter* CIP resistance in some regions, with rates exceeding 80% [20,21]. ERY resistance in *Campylobacter* infections is relatively low compared to other antibiotics, ranging to 1.8% in Australia [22] and 3.3% in the United States [23] from 75.3% in China [24]. ERY resistance in Latin American countries was generally low but present [25], supporting our results. This resistance is primarily driven by genetic mutations and is exacerbated by factors such as international travel and widespread fluoroquinolone/macrolide use in agriculture, animal, and human medicine [20,21,22,23,24,26,27,28].

Notably, all ERY-resistant isolates in our cohort correlated with unfavorable IQ outcomes, suggesting that even modest resistance rates may compromise clinical efficacy. This trend is particularly concerning given macrolides’ status as first-line agents for severe campylobacteriosis in children [28], and without stewardship interventions, the erosion of macrolide efficacy could leave few alternatives for pediatric populations. Our cohort exemplifies the vulnerability of young children to *Campylobacter* infections, compounded by limited therapeutic options. Tetracyclines—a potential alternative—are contraindicated in children <8 years, while azithromycin’s rising resistance leaves FZD as a pragmatic, albeit off-label, choice in some countries. This contrasts with international guidelines that do not endorse FZD for campylobacteriosis due to insufficient efficacy and safety data [29,30,31]. However, in LMICs like Peru and Brazil, where resistance surveillance is sparse, and antibiotic access is fragmented, FZD’s affordability and availability render it a de facto option. This disparity calls for a reevaluation of global guidelines to reflect regional resistance patterns.

### 3.3. Limitations

First, its single-center design may limit generalizability, though the high resistance rates observed align with regional surveillance data [25,26]. Second, our reliance on regression-derived MICs, though validated against CLSI standards [18,32], requires confirmation via gold-standard methods. Third, our MIC analyses further revealed significant discrepancies between CLSI and EUCAST interpretations for CIP (*p* = 0.022 vs. p = 0.008), underscoring the urgency for harmonized guidelines. Fourth, it is possible that dietary variation may influence the effect of FZD in vivo, as this factor can affect the microbial environment and therapeutic outcomes [33]. Future studies are needed to confirm our in vitro findings, which have demonstrated a positive outcome for FZD. Finally, the lack of clinical outcome data (e.g., symptom resolution timelines) precludes a direct correlation between in vitro susceptibility and therapeutic success. Prospective studies should prioritize FZD’s pharmacokinetic/pharmacodynamic profile in children [34], particularly dose optimization, to bridge the “intermediate” susceptibility gap.

## 4. Methods

### 4.1. Study Design and Ethical Approval

This prospective study was conducted in the Stool Culture Unit of the Microbiology Department at the Hospital Nacional Docente Madre Niño San Bartolomé in Lima, Peru, in two phases: Phase 1 (November 2012 to February 2013) and Phase 2 (December 2014 to July 2015). The study protocol was approved by the Ethics and Research Committee of the hospital’s Research Support Department. The number of participants in Phase 1 and Phase 2 was 88 and 106 samples, respectively. This study followed the recommendations of the STROBE guidelines [35].

### 4.2. Sample Collection, Inclusion Criteria, and Variables

Stool samples were collected from infants and children aged 0 to 24 months, with prior informed consent obtained from parents or guardians. Samples were collected using wet swabs for neonates or stored in sterile plastic containers or vials by nurses or caregivers [36,37]. Samples exhibiting signs of inflammatory reaction (e.g., presence of mucus or leukocytes) were referred from outpatient or parasitology departments for stool culture analysis.

We excluded patients older than 2 years, solid stool samples or those not meeting minimum quality requirements, and samples that were improperly preserved, contaminated, or exceeding 6 h since collection. The major inclusion criterion was the isolation of *Campylobacter*. The sample size was calculated using EPIDAT v.4.1 (Xunta de Galicia, Santiago, Spain), considering a power of 95%, heterogeneity of 50%, and an accuracy of 0.04, obtaining a sample size of 200 patients.

The primary study variable was the therapeutic efficacy of FZD against *Campylobacter*, while secondary variables were the antimicrobial susceptibility outcomes in FZD, ERY, and CIP.

### 4.3. Bacterial Isolation and Culture

Stool samples were coded using the hospital’s internal system (SIGOS) and evaluated for inflammatory markers (e.g., ≥7 leukocytes per field, positive mucus test). Samples were preserved in BBL Culture Swab™ Plus (BD, Pont-de-Claix, France) or Amies agar (Oxoid, Hampshire, UK) at room temperature [38]. Samples with compromised preservation buffers were excluded.

Samples were processed within 2 h of collection to prevent overgrowth of normal flora, which could obscure or destroy enteropathogens [39]. Culture media were pre-incubated at 37 ± 1 °C for 2 h to remove excess condensation. Samples were homogenized by inversion and circular agitation before inoculation onto seven selective media: Hektoen Agar, TCBS Agar, SMAC Agar, Salmonella/Shigella Agar, XLD Agar, Karmali Agar, and Sheep Blood Agar 5% with Grid GN-6 Metricel^®^ membrane filters 0.45 μm, 47 mm (PALL, Port Washington, NY, USA) (all of media culture of Merck, Darmstadt, Germany). *Campylobacter* isolation exclusively utilized Karmali agar and 5% sheep blood agar. The other media facilitated the isolation of medically relevant *Enterobacteriaceae*, following the hospital’s standardized protocol. Karmali Agar and Sheep Blood Agar were incubated at 42 ± 1 °C for 72 h under microaerophilic conditions (5% O_2_, 10% CO_2_, 85% N_2_) [8,40,41]. *Campylobacter jejuni* ATCC 33291 was used as a quality control strain under identical conditions.

### 4.4. Bacterial Identification

Plates were examined at 12, 48, and 72 h under a high-intensity light source to identify colonies with characteristic morphologies: (a) gray, flat, irregular colonies; (b) gray, round, convex, and shiny colonies with defined edges; (c) gray-greenish, moist wet colonies with irregular edges [8,42].

Suspect *Campylobacter* colonies were confirmed by Gram staining, revealing Gram-negative, spiral, or curved “gull-wing” shaped bacilli. Species differentiation was performed using the hippurate hydrolysis test and visualization of Ruhemann’s purple reaction [43]. Cultures with no growth after 96 h were considered negative. All confirmed *Campylobacter* isolates were stored at ≤40 ± 2 °C for further evaluation by the National Health Institute (External Quality Control Program).

### 4.5. Antibiotic Susceptibility Testing

Antibiotic susceptibility was determined using the Kirby–Bauer disk diffusion method. The following antibiotics were tested: erythromycin (ERY), ciprofloxacin (CIP), and furazolidone (FZD) (all of Oxoid Ltd., UK). Inocula were prepared in Mueller–Hinton Broth adjusted to 0.5 McFarland standard (~1.5 × 10^8^ CFU/mL) and plated on Mueller–Hinton Agar (MHA) supplemented with 5% sheep’s blood. Plates were incubated under microaerophilic conditions at 42 ± 1 °C for 48 h.

Breakpoints for ERY and CIP were interpreted according to CLSI guidelines (M45 and M100S) [18,32]. For FZD, nitrofurantoin breakpoints for *Enterobacteriaceae* were applied. Additionally, EUCAST zone diameter breakpoints were used for comparative evaluation (FZD for *Enterobacteriaceae* breakpoint) [19].

### 4.6. Determination of Minimum Inhibitory Concentration (MIC)

MIC values were determined by extrapolating inhibition zone diameters from disk diffusion assays onto regression curves. Regression curves were generated using critical cut-off points for resistance and susceptibility (expressed in mm and μg/mL) based on CLSI M45 and M100S guidelines and the exponential regression formula [44,45].

### 4.7. Inhibitory Quotient (IQ) Determination

The inhibitory quotient (IQ) was calculated using pharmacodynamic principles, dividing the theoretical antibiotic concentration by the MIC value [45,46,47,48]. IQ cut-off limits for fluoroquinolones, macrolides, and nitrofurans were established based on therapeutic response ratios. Results were interpreted using predefined IQ thresholds for each antibiotic.

### 4.8. Statistical Analysis

Antimicrobial susceptibility, MIC, and IQ data were reviewed by two authors before statistical processing to avoid confounding factors (J.M-S. and R.S-H.). Descriptive statistics were used to analyze primary study variables. Differences between CLSI and EUCAST breakpoints were assessed using Chi-square and Fisher’s exact tests, while confidence intervals for proportions were calculated using the Wilson Score Interval method [49]. Conventional antibiotic susceptibility interpretations and IQ analyses were performed using IBM SPSS v21.0 (Armonk, NY, USA). A *p*-value < 0.05 and a confidence interval of 95% (95% CI) were considered statistically significant.

### 4.9. Ethical Statement

This study has adhered to ethical guidelines and has been approved by the Hospital IRB (Exp. N145-20445-2013-HONADOMANISB-CEI).

## 5. Conclusions

This study positions FZD as a promising candidate for managing pediatric *Campylobacter* infections in high-resistance settings. Its sustained efficacy against MDR strains and low resistance propensity offers a lifeline for regions grappling with post-antibiotic realities. However, transitioning FZD from empirical use to evidence-based therapy requires robust clinical trials, updated breakpoints, and harmonized global guidelines. This research introduces FZD as a novel therapeutic option for pediatric *Campylobacter* infections, particularly in areas facing high antibiotic resistance as low- and middle-income countries. Its effectiveness against multidrug-resistant strains and minimal resistance development highlights its potential role in addressing the challenges of post-antibiotic treatment landscapes, requiring further clinical validation and standardized protocols.

## Figures and Tables

**Figure 1 antibiotics-14-00636-f001:**
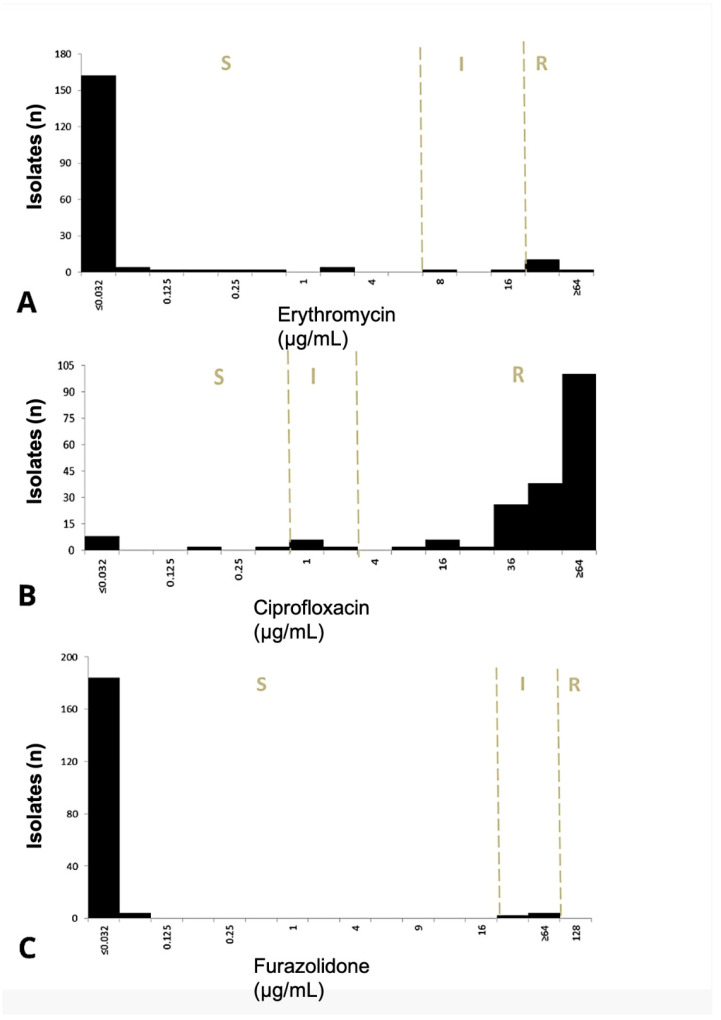
Antimicrobial MICs among *Campylobacter* isolates from pediatric patients. (**A**) Erythromycin. (**B**) Ciprofloxacin. (**C**) Furazolidone CLSI breakpoints (dashed lines) are shown. S: susceptible; I: intermediate; R: resistant. N = 194.

**Table 1 antibiotics-14-00636-t001:** Baseline finding of the therapeutic efficacy in vitro of furazolidone against erythromycin and ciprofloxacin for antimicrobial susceptibility testing of *Campylobacter* spp., *C. jejuni*, and *C. non-jejuni* in Peruvian pediatric patients. N = 194. Data in N%.

	CLSI (mm)									EUCAST (mm)						MIC (μg/mL) ^§^								
	ERY			CIP			FZD			ERY *		CIP		FZD		ERY			CIP			FZD		
	R	I	S	R	I	S	R	I	S	R	S	R	S	R	S	R	I	S	R	I	S	R	I	S
** *C. jejuni* **	2	0	28	26	0	4	0	4	26	4	26	28	2	0	30	2	0	28	26	0	4	0	4	26
** *C.no-jejuni* **	8	2	66	62	8	6	0	2	74	16	60	72	4	0	76	8	2	66	62	8	6	0	2	74
***C.* spp.**	2	0	86	86	0	2	0	0	88	2	86	86	2	0	88	2	0	86	86	0	2	0	0	88
Total	12	2	180	174	8	12	0	6	188	22	172	186	8	0	194	12	2	180	174	8	12	0	6	188

* EUCAST’s Breakpoints of erythromycin for *C. jejuni* (R: ≤20 mm, S: ≥20 mm) and *C. coli* (R: ≤24 mm, S: ≥24 mm). ^§^ The MIC analysis was performed only with the CLSI’s breakpoints.

## Data Availability

Data are contained within the article.

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
