# Peer review of "In Vitro Therapeutic Efficacy of Furazolidone for Antimicrobial Susceptibility Testing on Campylobacter"

_antibiotics, 2025, doi:10.3390/antibiotics14070636_

Round 1
Reviewer 1 Report
Comments and Suggestions for Authors
The authors should consider the followings:
The authors should provide the study schedule, to illustrate the arms and no. and reasons of inclusion/exclusion.
As for 2.8. Statistical Analysis, the authors should describe how they treat the missing data. And list any confounding factors clearly.
The authors should justify for the sample size estimation. The authors should estimate the sample size from the similar studies where possible, "The sample size was calculated using EPIDAT v.4.1 (Xunta de Galicia, Spain) considering a power of 95%, heterogeneity of 50% and an accuracy of 0.04 obtaining a sample size of 200 patient". What prior data made the authors decided on 0.04 accuracy?
The authors should pre-define the primary and secondary endpoint(s) clearly.
Limitations of the study should be discussed.
The authors should explain how they decided the dose ranges of the study and the choice of drugs.
Dietary variation and storage of the stool (before submission to lab) should be monitored and documented carefully, since these factors may affect the microbial environment and the study results.
The novelty of the study should be clearly stated in the abstract and/or conclusion.
The authors may include more relevant and updated references to the study references.
Comments on the Quality of English LanguageQuality of English may need further improvement.
Author Response
The authors should consider the followings:
The authors should provide the study schedule, to illustrate the arms and no. and reasons of inclusion/exclusion.
RESPONSE: This part has been improved
As for 2.8. Statistical Analysis, the authors should describe how they treat the missing data. And list any confounding factors clearly.
RESPONSE: No missing data were reported and confounding data were added.
The authors should justify for the sample size estimation. The authors should estimate the sample size from the similar studies where possible, "The sample size was calculated using EPIDAT v.4.1 (Xunta de Galicia, Spain) considering a power of 95%, heterogeneity of 50% and an accuracy of 0.04 obtaining a sample size of 200 patient". What prior data made the authors decided on 0.04 accuracy?
RESPONSE: A 40% accuracy rate was suggested by the hospital's IRB, given that the number of Campylobacter samples in children is limited, and no previous studies on FZD have been conducted in the region. In an attempt to reduce the error rate, the study's error rate was considered. In this same sense, the statistical power of the study was increased in order to obtain greater certainty in the sample calculation.
The authors should pre-define the primary and secondary endpoint(s) clearly.
RESPONSE: It has been included in line 94-97
Limitations of the study should be discussed.
RESPONSE: The limitations are in 4.3. of the discussion (lines 275-285)
The authors should explain how they decided the dose ranges of the study and the choice of drugs.
RESPONSE: It has been included in line 133-136. Antibiotic selection criteria were followed according to the recommendations of the CLSI and EUCAST guidelines.
Dietary variation and storage of the stool (before submission to lab) should be monitored and documented carefully, since these factors may affect the microbial environment and the study results.
RESPONSE: Dietary variation has been included in the limitations section. In this study, we can confirm that sample storage did not represent a limitation for the in vitro analysis of microbial susceptibility, as scientific rigor was maintained to avoid this bias.
The novelty of the study should be clearly stated in the abstract and/or conclusion.
RESPONSE: The novelty of the study has been added to the conclusion.
The authors may include more relevant and updated references to the study references.
RESPONSE: Additional current references have been included
Comments on the Quality of English Language: Quality of English may need further improvement.
RESPONSE: The English text has been improved, please review.
Reviewer 2 Report
Comments and Suggestions for Authors
This manuscript presents a two-phase study evaluating the emergence of antibiotic resistance to FZD, CIP, and ERY in Campylobacter isolates from a high-risk pediatric cohort in Lima, Peru. The authors combine Kirby-Bauer disk diffusion, MIC testing, and IQ analysis to assess the clinical relevance of the resistance profiles. The experimental design is appropriate, the statistical methods are correctly applied, and the results are interesting and potentially helpful in guiding physicians in the region on drug use and administration. However, the manuscript appears surprisingly simple, especially with respect to the interpretation and presentation of the data and figures. Nevertheless, conceding the challenges for clinical sample collection, I recommend revisions of the manuscript.
Major point:
In addition to directly presenting their conclusions, could the authors provide more figures or tables to support their claims?
Minor point:
Line 152-153: Does this mean that in the first study phase, the isolates that were resistant to ERY were also resistant to CIP? Does author have any comments? What about the second study phase?
Line 154: Does author have data that supports the statement "All ERY- and CIP-resistant strains correlated with unfavorable therapeutic responses"?
Line 179: MIC: 9≤36?
Line 202: Why can't the data be shown? It would be helpful to provide all of the data that isn't shown as additional tables. I also suggest providing all of the raw data from the entire manuscript as supplementary tables.
Author Response
This manuscript presents a two-phase study evaluating the emergence of antibiotic resistance to FZD, CIP, and ERY in Campylobacter isolates from a high-risk pediatric cohort in Lima, Peru. The authors combine Kirby-Bauer disk diffusion, MIC testing, and IQ analysis to assess the clinical relevance of the resistance profiles. The experimental design is appropriate, the statistical methods are correctly applied, and the results are interesting and potentially helpful in guiding physicians in the region on drug use and administration. However, the manuscript appears surprisingly simple, especially with respect to the interpretation and presentation of the data and figures. Nevertheless, conceding the challenges for clinical sample collection, I recommend revisions of the manuscript.
Major point:
In addition to directly presenting their conclusions, could the authors provide more figures or tables to support their claims?
RESPONSE: Our results are presented as follows:
- Table of therapeutic efficacy for the three drugs and Campylobacter species
- Figure 1 shows the MICs of the drugs
The study objectives are framed in both results, so we believe it is sufficient to answer the study's research question.
Minor point:
Line 152-153: Does this mean that in the first study phase, the isolates that were resistant to ERY were also resistant to CIP? Does author have any comments? What about the second study phase?
RESPONSE: Only a few isolates were resistant to both drugs; this information from phase 2 has been included.
Line 154: Does author have data that supports the statement "All ERY- and CIP-resistant strains correlated with unfavorable therapeutic responses"?
RESPONSE: The data are mentioned in 3.3 and can be verified in Figure 1 using MICs. These indicate that FZD has an adequate therapeutic response.
Line 179: MIC: 9≤36?
RESPONSE: It has been reviewed and modified.
Line 202: Why can't the data be shown? It would be helpful to provide all of the data that isn't shown as additional tables. I also suggest providing all of the raw data from the entire manuscript as supplementary tables.
RESPONSE: The data were not disclosed at the request of the hospital's IRB. Because these data (including the raw data) are part of a research project, the hospital withholds the information until publication. After publication, secondary analyses can be performed with the available information.
Reviewer 3 Report
Comments and Suggestions for Authors
In lines 162 to 163, the authors excluded 485 (82%) samples and only analyzed 106 (18%) samples. Could the authors explain why they excluded most of the samples and only analyzed a few of the samples? Even though the authors mentioned non-compliance with inclusion criteria as the reason, could the authors provide the details of how they selected to analyze the samples?
In lines 178 to 179, could the authors provide the MIC range for the intermediate category?
In Figure 1, could the authors give an explanation why the percentages of isolates are over 100%?
In Figure A.2., could the authors clarify what are in tubes 5 and 6 and where are tubes 1 and 2?
Author Response
In lines 162 to 163, the authors excluded 485 (82%) samples and only analyzed 106 (18%) samples. Could the authors explain why they excluded most of the samples and only analyzed a few of the samples? Even though the authors mentioned non-compliance with inclusion criteria as the reason, could the authors provide the details of how they selected to analyze the samples?
RESPONSE: The text has been modified to improve its understanding.
In lines 178 to 179, could the authors provide the MIC range for the intermediate category?
RESPONSE: We have added an explanatory line on lines 88-89, and the results text has been modified.
In Figure 1, could the authors give an explanation why the percentages of isolates are over 100%?
RESPONSE: The data are not in %, but numerical. This has already been corrected in the figure.
In Figure A.2., could the authors clarify what are in tubes 5 and 6 and where are tubes 1 and 2?
RESPONSE: Tubes 1 and 2 are on the left side (the tubes are not visible), and all the uncolored tubes are negative. This has been added to the text.
Reviewer 4 Report
Comments and Suggestions for Authors
The manuscript presents an intriguing study on the in vitro therapeutic efficacy of furazolidone for antimicrobial susceptibility testing on Campylobacter. This research broadens my understanding of the topic and provides valuable insights into alternative antimicrobial agents against Campylobacter infections.
However, I have a few concerns that should be addressed to enhance the clarity and accuracy of the manuscript:
Title Formatting: The term "in vitro" in the title should be italicized to conform with standard scientific writing conventions.
Abstract Formatting: The abbreviation "C. jejuni" should be italicized in the abstract to align with proper taxonomic nomenclature.
Introduction Section: The full name "Campylobacter jejuni subsp. jejuni (C. jejuni)" and other species, such as "C. lari," should be italicized. I recommend carefully reviewing the manuscript to ensure that all bacterial names are correctly formatted.
Methodology – Selective Media: The manuscript mentions the use of seven selective media for Campylobacter cultivation. However, some of these media are not typically suitable for Campylobacter growth. A detailed explanation should be provided to justify the use of multiple selective media in this experiment.
Methodology – Taxonomic Nomenclature: The term "Enterobacteriaceae" should be italicized to adhere to scientific conventions.
Results – Figure 1: The font size of the labels on both the X- and Y-axes should be increased to improve readability.
Addressing these points will strengthen the manuscript by improving clarity, formatting consistency, and methodological justification.
Author Response
The manuscript presents an intriguing study on the in vitro therapeutic efficacy of furazolidone for antimicrobial susceptibility testing on Campylobacter. This research broadens my understanding of the topic and provides valuable insights into alternative antimicrobial agents against Campylobacter infections.However, I have a few concerns that should be addressed to enhance the clarity and accuracy of the manuscript:
Title Formatting: The term "in vitro" in the title should be italicized to conform with standard scientific writing conventions.
RESPONSE: We added italics in the tittle term.
Abstract Formatting: The abbreviation "C. jejuni" should be italicized in the abstract to align with proper taxonomic nomenclature.
RESPONSE: We added italics in the term.
Introduction Section: The full name "Campylobacter jejuni subsp. jejuni (C. jejuni)" and other species, such as "C. lari," should be italicized. I recommend carefully reviewing the manuscript to ensure that all bacterial names are correctly formatted.
RESPONSE: We revised all terms in the text, we made changes as suggestions.
Methodology – Selective Media: The manuscript mentions the use of seven selective media for Campylobacter cultivation. However, some of these media are not typically suitable for Campylobacter growth. A detailed explanation should be provided to justify the use of multiple selective media in this experiment.
RESPONSE: The text has been modified for better explanation.
Methodology – Taxonomic Nomenclature: The term "Enterobacteriaceae" should be italicized to adhere to scientific conventions.
RESPONSE: We added italics in the term.
Results – Figure 1: The font size of the labels on both the X- and Y-axes should be increased to improve readability.
RESPONSE: The figure has been revised and improved
Round 2
Reviewer 2 Report
Comments and Suggestions for Authors
The authors have adequately addressed all previous concerns. I am satisfied with the revisions and recommend acceptance for publication.
Reviewer 3 Report
Comments and Suggestions for Authors
I have no further comments.